## PROCEEDINGS A

mechanics, relativity, mathematical physics

Foucault pendulum, parametric excitation, nonlinear dynamics, Lense–Thirring precession, general relativity

**Author for correspondence:**
Matthew P. Cartmell
e-mail: matthew.cartmell@strath.ac.uk

# On the modelling and testing of a laboratory-scale Foucault pendulum as a precursor for the design of a high-performance measurement instrument

Matthew P. Cartmell[1], James E. Faller[2,3], Nicholas A. Lockerbie[3,4] and Eva Handous[5]

[1]Aerospace Centre of Excellence, Department of Mechanical and Aerospace Engineering, University of Strathclyde, Glasgow G1 1XJ, UK
[2]JILA, University of Colorado, Boulder, CO 80309, USA
[3]Institute for Gravitational Research, School of Physics and Astronomy, College of Science and Engineering, University of Glasgow, Glasgow G12 8QQ, UK
[4]Emeritus Professor of Physics, University of Strathclyde, Glasgow G4 0NG, UK
[5]Ecole Nationale Supérieure d'Arts et Métiers, Campus de Lille, 8 Boulevard Louis XIV, 59000 Lille, France

MPC, 0000-0002-3982-6315

An integrated study is presented on the dynamic modelling and experimental testing of a mid-length Foucault pendulum with the aim of confirming insights from the literature on the reliable operation of this device and setting markers for future research in which the pendulum may be used for the measurement of relativistic effects due to terrestrial gravity. A tractable nonlinear mathematical model is derived for the dynamics of a practical laboratory Foucault pendulum and its performance with and without parametric excitation, and with coupling to long-axis torsion is investigated numerically for different geographical locations. An experimental pendulum is also tested, with and without parametric excitation, and it is shown that the model closely

predicts the general precessional performance of the pendulum, for the case of applied parametric excitation of the length, when responding to the Newtonian rotation of the Earth. Many of the principal inherent performance limitations of Foucault pendulums from the literature have been confirmed and a general prescription for design is evolved, placing the beneficial effect of principal parametric resonance of this inherently nonlinear system in a central mitigating position, along with other assistive means of response moderation such as excitational phase control through electromagnetic pushing, enclosure, and the minimization of seismic and EMC noise. It is also shown, through a supporting analysis and calculation, that although the terrestrial measurement of the Lense–Thirring (LT) precession by means of a Foucault pendulum is certainly still within the realms of possibility, there remains a very challenging increase in resolution capability required, in the order of $2 \times 10^9$ to be sure of reliable detection, notwithstanding the removal of extraneous motions and interferences. This study sets the scene for a further investigation in the very near future in which these challenges are to be met, so that a new assault can be made on the terrestrial measurement of LT precession.

## 1. Introduction

Léon Foucault was a notable French physicist, who, apart from measuring the speed of light and discovering eddy currents during his illustrious career, proposed a striking experiment in 1851 to show visually the rotation of the Earth in a direct manner by means of a carefully suspended long pendulum. The installation of a Foucault Pendulum requires great care and precision if one wants to observe the real precession generated by the rotation of the Earth beneath the laboratory, without other spurious effects intruding and ultimately dominating the motion of the pendulum. Since its inception, the Foucault pendulum has received a large amount of attention and many of the potential design problems, which are now known to be inherent to this system, have been investigated in detail and a multitude of mitigating solutions have been proposed during more than a century of international research since the time of Foucault. The aim of the work discussed within this paper is twofold. Firstly, we propose a straightforward but representative mathematical model of the Foucault pendulum including parametric excitation of the length in the form of vertical support motion, and we offer a numerical study of the predicted responses for a number of parameter cases and geographical locations. This model clearly complements those available in the literature and also provides a useful level of transparency of derivation, which, we believe, provides an additional aid to understanding. The other intention has been to undertake a novel laboratory test programme, based on our interpretation of the extant literature, and to demonstrate a pragmatic but effective level of design optimization. In order to achieve this, we have attempted to design and build a prototype that is technically simple and also possesses the important physical symmetries that are necessary. In addition, the pendulum is driven into principal parametric resonance to try to minimize a very well-known difficulty which is when the initially planar motion of the pendulum degenerates into ellipticity, this being associated with highly undesirable frequency anisotropy effects which have been widely reported within the experimental literature over the years. The design presented in this paper shows that a parametrically excited Foucault pendulum of medium length can indeed be built for a reasonable outlay and installed in an average laboratory for long-term tests initially of the classical Earth rotation problem. Ultimately, our goal is to develop the design into a more sensitive, and larger-scale instrument capable of resolving relativistic effects within a terrestrial laboratory in a northerly location.

A literature review has shown that *ellipticity* is mainly caused by structural asymmetries either due to defects in design, or manufacture, or both, which result in different periods of each axis of the resulting ellipse, noting [1] that *every Foucault Pendulum, no matter how carefully constructed to avoid asymmetries in its suspension, and no matter how carefully 'launched' to make ellipsoidal area A*

*as small as reasonably achievable will, over time, acquire an intrinsic precession* $\Omega$ *that can easily grow to overwhelm the Coriolis-force-induced Foucault precession* $\Omega_F$. However, to prevent this elliptical motion of the pendulum, it has to remain as long as possible [2,3] and with the highest mass allowed [2], noting that the original pendulum was 67 m in length and the wire was steel which had been 'strained somewhat beyond its elastic limit' [2]. It is generally accepted that the support of the pendulum must be perfectly symmetrical and a ball joint pivot or a double bifilar suspension are preferable to a double knife edge [2]. Annealing the wire may possibly permit the reduction in ellipticity but embrittlement follows such a treatment [2]. Torsionless wires of silk or cotton were tested, but the use of such materials apparently does little to remove ellipticity [4]. An electromagnetic drive using coils, coupled with a Charron ring, was shown to be effective [3,5] for short pendulums of a few metres. The Charron ring is a fixed metallic annulus surrounding the wire at about 1/10th of the way down from the top, making mechanical contact at large amplitudes and in doing so encouraging the pendulum to reduce its ellipticity of response. A specialized electromagnetic drive system was suggested by Crane [3] in which the bob carried a small vertically orientated permanent magnet protruding from the bottom of the spherical bob and this passed over a circular coil which gave the bob two outward and two inward pushes per cycle. This was intended to average out any asymmetry in the drive in the case where the amplitude was constant and where the minor diameter of the ellipse remained small compared to the separation of the poles of the two magnets. Priest & Pechan [6] re-considered the drive required to maintain Foucault pendulum motion and suggested an electronic timing circuit and an updated design for the mount and collar with a drive electromagnet and feedback sense coil to provide an indication of the proximity of the bob to the drive electromagnet, and to control the current. Salva *et al.* [7] studied ellipticity due to support asymmetry and used a similar electronic drive system to that of Crane [3] in which the bob was accelerated towards the centre. They used an electromagnetic brake to keep the ellipticity small, but noted it still persisted to some extent. Their system was tested with two pendulums: the first of length 2835 mm and the second of length 4970 mm. The bob was spherical of 12.5 kg and the suspension wires were, respectively, a 0.92 mm diameter steel wire and a 0.92 mm diameter USA Diamond Brand chord piano wire.

Braginsky *et al.* [8] proposed an extraordinarily ambitious experiment in 1984 in which a Foucault pendulum and an astronomical telescope, colocated at the South Pole, might be used together to measure the angle between the principal axis of the swing and the azimuth of a reference star such as Canopus to detect eventually the tiny relativistic effect due to the Lense–Thirring (LT) precession, and in this way to obtain a terrestrial measurement that is directly associated with general relativity. Their paper highlighted in considerable detail the various *dangerous sources of experimental error and the methods for circumventing them*, listed as follows: (i) *Magnetic forces* from the Earth's magnetic field interacting with the natural charge on the pendulum, noting that charge can be minimized by coating the bob and the wire, in the form of fused quartz fibre, with a thin layer of metal, (ii) *Frictional damping* through frictional anisotropy, for which a component will have to be subtracted from the results, (iii) the so-called *Pippard precession* as a consequence of using a support fixed to the Earth which introduces a related spin angular momentum whose direction changes as the pendulum swings, generating a small precession which would also have to be subtracted from the results, (iv) *Position-dependent forces* emanating from gravity gradient or even light beam pressure generating a growing ellipticity and a possible associated precession, (v) *Frequency anisotropy* due to the finiteness of the amplitude of the swing, giving rise to different periods in the orthogonal directions, and pronounced ellipticity in the response, as previously discussed and noting that an electrostatic correction system was also suggested for this, (vi) *Seismic noise* for which substantial anti-seismic isolation will be needed. Acknowledging this particularly significant problem, Braginsky *et al.* [8] ingeniously suggested the use of two pendulums attached to the same support but running out of phase and so cancelling each other's seismically induced precessions, (vii) *Atmospheric refraction* whereby variations in the measured position of the reference star will be caused by changes in azimuthal atmospheric refraction both near the telescope and far from it. South Pole location would help, and effects could be reduced further by tracking two stars on opposite sides of the sky.

They observed that a further correction could be made by two-colour refractometry, (viii) *Distortion of the telescope* which manifests as any instability in the azimuthal optics due to gravitational stresses, temperature fluctuations and ageing. They stated that the thermal stability across the telescope would need to be maintained to around 0.01 K, (ix) *Tilt of the telescope* due to the light feed mirror tilt affecting the apparent azimuthal position of the star. Feedback control would be required to hold the light feed mirror perfectly steady. Interestingly, all these issues were said to be technically addressable in 1984 [8], and we return to this particular application later in the discussion of §6. Gusev *et al.* [9] investigated effects of viscous friction, seismic noise and dynamic instability and modelled the anisotropy of the suspension, and then in 2002, Pascual-Sánchez [10] returned to the Braginsky *et al.* proposal for the measurement of LT precession, referring to this enhanced proposal as the TELEPENSOUTH experiment. In response to the list of error sources proposed by Braginsky *et al.* [8], Pascual-Sánchez [10] suggested using a sapphire fibre of diameter 0.1 mm and mass of 0.1 kg, a long thin and dense mass for the bob (tungsten suggested) and a length to amplitude ratio of 40. Iorio [11] incorporated many of the recommendations of [10] in his interesting description of a desk-bound version of the Braginsky *et al.* [8] experiment.

Pippard [12] built in 1988 what seems to have been the first parametrically excited Foucault pendulum using length excitation and showed that parametric excitation can act strongly against ellipticity, while increasing the gain of the Foucault pendulum as a measurement device, and that it also opposes the inherent damping of the pendulum. Pippard undertook a thorough analysis of the role of principal parametric resonance in the Foucault pendulum and although he was generally rather pessimistic about the probable success of using one as an instrument for measuring relativistic precessional motions, he nevertheless made a brief mathematical statement of the measurement of LT precession. Tungsten was recommended for the wire because of low creep properties and Pippard derived a useful relationship between bob mass and wire diameter for controlling the strain in tungsten in order to guarantee minimal creep over time. Reference was also made to the design of Mastner *et al.* [13] in which damping was linearized through the evacuation of a chamber surrounding the pendulum and by means of the synchronized rocking of a conductive eddy current damper plate at a precise amplitude [12].

Salas & Flores [14] contributed importantly to Foucault pendulum instrumentation in 2004 by proposing an imaging system which would offer the following functionality (i) detection of the bob under changes in the background scenario, (ii) computation of the bob's trajectory by fitting an ellipse to the set of observed positions, and (iii) estimation of the noise in the predominant direction of motion and reduction using a Kalman filter. Interestingly, their pendulum was deliberately designed to be of relatively low performance, with no special attention to be paid to the collar and pivoting system, so elliptical motion was expected to be a certainty and because of this, the system provided a particularly compelling test-bed for their compensatory instrumentation.

Stanovnik [15] quite correctly pointed out that two-dimensional modelling cannot directly account for the vertical component of bob motion, which grows with swing amplitude, and also when taking the elasticity of the string into account should that be significant in the design to be studied. In such cases, the use of a three-dimensional model was advocated in order to build in string dilatation due to bob mass and an additional elastic variation in string length during motion. According to this author, the long-term precession period of a two-dimensional Foucault pendulum model may not necessarily equate to the rotation period of the Earth as the precession period of a Foucault pendulum can be dependent on both the rotation of the Earth and the elastic properties of the suspension string, thus requiring a three-dimensional nonlinear dynamic model. The consequence for an experimental system is that the string elasticity can be significant, and needs to be properly considered.

In the important recent paper by Schumacher & Tarbet [1], it was claimed that ellipsoidal precession can be removed electromagnetically and that the method that they proposed for this would be insensitive to the size and direction of the perturbation forces leading to ellipsoidal motion. They used a short 3 m pendulum that could be 'pushed' in a controllable manner to

make the Foucault precession dominant. They emphasized that although the longer Foucault pendulum will generally show less susceptibility to the ellipticity problems that submerge the desired effect, this geometrical feature alone will not eradicate it completely. Their starting point was the desirability of building shorter Foucault pendulums in which the accrued ellipsoidal motion might be minimized, and then to compensate effectively for the irreducible amount of ellipsoidal precession that remains. Their method used the idea that pushing the bob away from the origin as it passes, rather than either pulling it in or alternately pulling and pushing it, acts in a way that counters the unwanted intrinsic ellipsoidal procession. They explained that when the pendulum is at its extremum (at $x = \pm a$, where $a$ is the semi-major axis), its motion would be entirely transverse with momentum $m\dot{y}$ being maximized. From this, they showed that preferential damping of this component of the motion would reduce the unwanted ellipsoidal excursions, and they noted that the mechanical Charron ring is one simple way of achieving this. They proposed a more modern eddy current damping method, but maintained that none of these methods will stop ellipsoidal motion completely, hence the need for their additional active approach. The principal contribution of [1] is a special electromagnetic drive that develops a magnetic push not only to compensate for the dissipative losses but also for the unwanted ellipsoidal precession. The authors stated four physical aspects of the design that can lead to ellipsoidal motion: (i) significant internal stresses or other imperfections in the wire of the pendulum, (ii) less than perfectly symmetrical suspension of the wire at its upper end, (iii) nearby ferrous objects that result in an asymmetric force on the drive electromagnet, (iv) a driving coil that is not sufficiently levelled and centred under the pendulum. They concluded that the last three can all be designed out, or minimized by good design, but the first cannot necessarily be eradicated. Their solution was to apply a separate perturbative force to nullify the intrinsic ellipticity. Schumacher & Tarbet [1] analysed their results in detail and the paper confirmed that their pushing-only solution worked very effectively.

Lacsny *et al.* [16] described a relatively simple experiment in which a short Foucault pendulum of only 2.85 m was used with a spark generator system by which the movement of the pendulum could be directly recorded by a spark burned trace on paper. The main contribution of this paper, other than the spark-burn motion detector, is the design of the anchorage. This used a hardened steel ball resting on a perfectly level (adjustable) polished aluminium plate that was cantilevered out from a solid wall support. The steel ball was at the end of a bar attached to a brass ring, and the pendulum was attached to the ring. The cylindrical bob was deflected from equilibrium to give the required displacement initial conditions and held there with a thread. This was burned at the point when the pendulum was to be set in motion.

A very recent paper on Foucault pendulum design by Plewes [17] discussed the possibilities for magnetic monitoring of a very short Foucault pendulum of less than 1 m in length. The author also listed the possible sources of inevitable ellipticity over repeated oscillations, as follows: radial asymmetry of system components, non-ideal alignment or levelling of components and non-uniform stresses in the pendulum wire and suspension. The author stated that *this erroneous precession can operate either to negate or augment the apparent Foucault precession from the Earth and therefore introduce significant errors.* Plewes [17] pointed out that both Pippard [12] and Olsson [18] studied ellipticity and that they showed the precession rate from this mechanism is well approximated by, $\Omega_{\text{error}} = 3\sqrt{g}\theta b/8l^{3/2}$, where $\theta$ is the pendulum swing angle (with respect to the local vertical), $b$ is the minor axis amplitude due to the ellipticity and $l$ is the pendulum length. This is certainly reasonable for small $\theta$. Short pendulums are said to be significantly more susceptible to this form of motion, now frequently referred to as Pippard precession. Relationships between minor axis displacement and target Pippard precession were given. Physical properties were also suggested for a practical design for a very short Foucault pendulum, notably the use of a pin vice for attachment, and spring-tempered, phosphate-coated, carbon steel wire of 0.812 mm diameter. Mounting of the pin vice was considered to be critical, and in particular accurate levelling. A neodymium permanent magnet was recommended within the bob for use with a circular detector coil and concentric drive coil. The drive coil should be very carefully designed in order to reduce any notably asymmetry in its field. A precision

magnetometer was recommended for measurement of precession and a circuit diagram was given for the timing and measurement electronics. The author also used a Charron ring, recommended as a supplementary device for mitigating ellipticity in short pendula.

Considering all these previous findings, it was decided that a new laboratory test pendulum should be built, principally by taking the following fundamental design criteria into account.

The literature is consistent that long pendulums with correspondingly high bob masses are less susceptible to ellipticity effects due to structural asymmetry than short systems with light bobs, so the longest pendulum possible within the available laboratory space was the first design decision. Suspension and pivoting has repeatedly been shown to be absolutely crucial and so the inherent symmetry and the simplicity of a high-performance spherical ball joint was chosen over bespoke needle or ball and plate joints, intricate gimbals, and complicated fluid or magnetic bearings. The literature has been very clear that ellipticity can be minimized by means of an appropriate electromagnetic 'pusher' system combined with symmetrical design and the use of quality materials, but parametric excitation is regarded as an effective supplementary choice because of the extremely strong response of nonlinear systems when subjected to principal parametric resonance [19,20]. It should be noted that this response is only actually bounded either mathematically or physically by the presence of nonlinearities within the system [20], and that the Foucault pendulum is suitably nonlinear. Parametric resonance has been found to remove ellipticity, overcome frictional and aerodynamic damping (if the parametric excitation amplitude is high enough and the resonance is strong) and to maintain a maximized response swing amplitude [12,20,21]. It is also very likely to remove the smaller effects of position-dependent force-induced precessions. In addition, the pendulum bob should not be of a shape that will exacerbate unwanted rocking, wobble, or torsional motions, and previous findings recommend a long, thin, cylindrical bob made from tungsten to meet these requirements. The material composition and geometry of the wire is highly critical and a conductive material is preferable in order to remove the unwanted electrostatic charge, and the wire should also be as longitudinally stiff as possible and generally substantial enough not to creep over time. As a consequence, an experimental design has been synthesized and this is discussed in detail in §4.

The novelty within the research presented here includes the derivation of a tractable generalized model for the terrestrial Foucault pendulum which includes nonlinear aerodynamic damping, parametric excitation and long-axis torsion, followed by a reasonably extensive numerical analysis showing both the capabilities and the shortcomings of the model, including a geographical context. In addition to this, a new experimental design for a mid-length Foucault pendulum is discussed in some detail, with literature contextualization, and tests of the unforced and parametrically excited pendulum are presented comparatively and critically. Finally, the paper concludes by re-visiting previous proposals for the terrestrial measurement of the gravitational LT precession, and offers a comparative re-calculation of that quantity with key results from the literature.

## 2. Mathematical model

In order to derive a mathematical model for the Newtonian dynamics of a terrestrial Foucault pendulum, we start by introducing a fundamental global frame of reference with its origin at the centre of the Earth, $EXYZ$, and then we identify a second frame of reference grounded at the location of the pendulum, defined by $pxyz$, as shown in figure 1. The Earth rotates about axis $EZ$, for which we define the associated unit vector $\bar{e}_Z$, and we denote the angular velocity of the Earth by $\bar{\Omega}$. The latitude of the pendulum location is given by $\phi$. The unit vectors for the local frame are $\bar{e}_x$, $\bar{e}_y$ and $\bar{e}_z$, and these can be identified in figure 2. We note that the plane defined by $pyz$ is considered to be coplanar with the plane defined by $EYZ$. The pendulum is of length $l$, where this is taken from the point of emergence of the wire from the pivot at some point on $pz$ to the centre of the bob at $B$. It swings through angle $\alpha$, and coordinates $x$ and $y$ are therefore defined by projecting down from $B$ to $A$, where $A$ is on the $pxy$ plane, all as shown in figure 2.

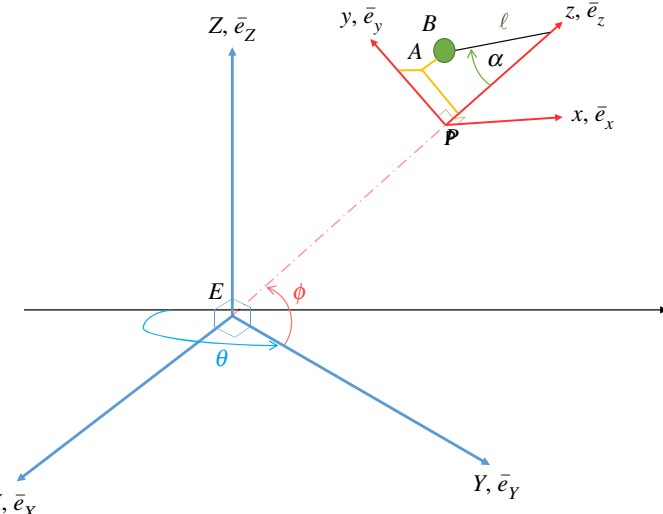

**Figure 1.** Earth-centred frame of reference *EXYZ* and local frame *pxyz*, showing pendulum deflected through $\alpha$ and located at a latitude defined by $\phi$. (Online version in colour.)

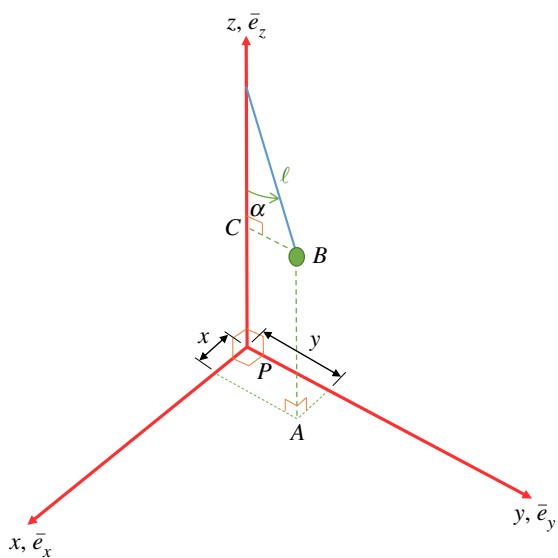

**Figure 2.** Local frame of reference *pxyz* rotated round for clarity of view, showing local Cartesian coordinates *x* and *y*. (Online version in colour.)

The derivation of the equations of motion is summarized in detail in the electronic supplementary material, file ESM1. The equations are as follows:

$$\ddot{x} + \eta|\dot{x}|\dot{x} - 2\dot{y}\Omega\sin\phi - x\Omega^2 + \frac{gx}{l\sqrt{1 - ((x^2 + y^2)/l^2)}} = 0 \tag{2.1}$$

and

$$\ddot{y} + \eta|\dot{y}|\dot{y} + 2\dot{x}\Omega\sin\phi - y\Omega^2\sin^2\phi + r\Omega^2\sin\phi\cos\phi + \frac{gy}{l\sqrt{1 - ((x^2 + y^2)/l^2)}} = 0, \tag{2.2}$$

where the quantity $\eta$ is the damping coefficient, defined in the electronic supplementary material.

## (a) Parametric excitation

As can be seen from equations (2.1) and (2.2), the system requires initial conditions to excite a transient response, but it is also perfectly possible to include a simple form of parametric excitation by adding a small modulation to the pendulum length in the following form, where the originally constant pendulum length $l$ now becomes $l(t)$

$$l(t) = l_0 + l_1 \mathrm{Cos}(\omega_1 t), \tag{2.3}$$

and where principal parametric resonance is defined by

$$\omega_1 = 2\omega_n + \epsilon\sigma. \tag{2.4}$$

The frequency $\omega_n$ is the undamped natural frequency of free vibration for the pendulum of nominal length $l_0$ and $\epsilon\sigma$ defines a small latitude, or detuning, around the perfectly resonant point. Given that a parametric resonance will generally destabilize a linear system from its null stability then appropriate combinations of excitation frequency $\omega_1$ (or detuning $\epsilon\sigma$) and excitation amplitude $l_1$ can push such a system through metastability and into the theoretically unstable zone where the response magnitude of a linear system is unbounded. In the case of a nonlinear system, as in the Foucault pendulum, the nonlinearities bound the response, so the system is not definitionally unstable and instead it responds vigorously up to a limit that is imposed by the dominant nonlinearity. The net effect of this is that as long as $\epsilon\sigma$ is very small then the principal parametric resonance will strongly amplify and maintain the transient response of the pendulum. The stability of a parametrically excited pendulum depends on whether or not it is linear or nonlinear. Obviously, any real-world system will be nonlinear to some extent but a close-to-linear parametrically excited system could be represented by the Mathieu–Hill form of differential equation, for which the solution at principal parametric resonance will be unbounded. This is very often depicted by means of zones of stability and instability defined by the so-called transition curve or the Strutt–Ince diagram, in which excitation amplitude and frequency are plotted, and where the curve depicts metastability, with transitions across it either to amplitude/frequency pairs that define stable (null) solutions or alternatively within the zone where the amplitude/frequency values define unbounded unstable solutions. The effect of damping such an oscillator is to pull the 'nose' of the transition curve up a little more from the frequency axis, so more excitation amplitude is required to overcome the damping so that the system can transition into instability. In the case of a nonlinear system, the nonlinearity bounds the instability to a finite value, so the 'unstable' response is actually a very large amplitude bounded value. If a notionally cubic nonlinearity dominates the parametrically excited system then there could be three bounded 'instabilities', with the system tending to gravitate to the upper one [20]. Basins of attraction could be calculated to home in exactly on specific cases. In the work reported here, the Foucault pendulum is nonlinear and so softening cubics naturally emerge. The numerical results in figure 3 show the most stable (upper) bounded solutions for the pendulum. A well-designed Foucault pendulum will tend to have a high $Q$ factor, and so the transient response should naturally persist for a long time. The benefits of using parametric excitation are threefold in the Foucault pendulum: (i) it amplifies the magnitude of the response and the potential benefits of this are explored further in the discussion in §6, (ii) it overcomes the natural decay due to damping, noting that we are modelling this as an aerodynamic effect here, to maintain the response at a high constant amplitude—as limited by the dominant nonlinearities, and finally (iii) it assists in the minimization of ellipticity by driving the pendulum to maintain the amplified form of the response starting from the chosen initial conditions. Using equations (2.1)–(2.4) inclusive allows us to examine initially the response of the Foucault pendulum to a parametric excitation of the length, as shown in figure 3. The amplification of the response from the initial displacement conditions (defined by the red dot) can be clearly seen. The nominal length of the pendulum $l_0$ is 8 m and the amplitude of the parametric length excitation $l_1$ is 0.075 m. The local acceleration due to gravity at Glasgow $g$ is $9.8156 \, \mathrm{m \, s^{-2}}$ and the angular rate of the Earth $\Omega$ is $7.2921150 \times 10^{-5} \, \mathrm{rad \, s^{-1}}$. The latitude of Glasgow $\phi$ is 0.9750 rad, and the local radius of the Earth,

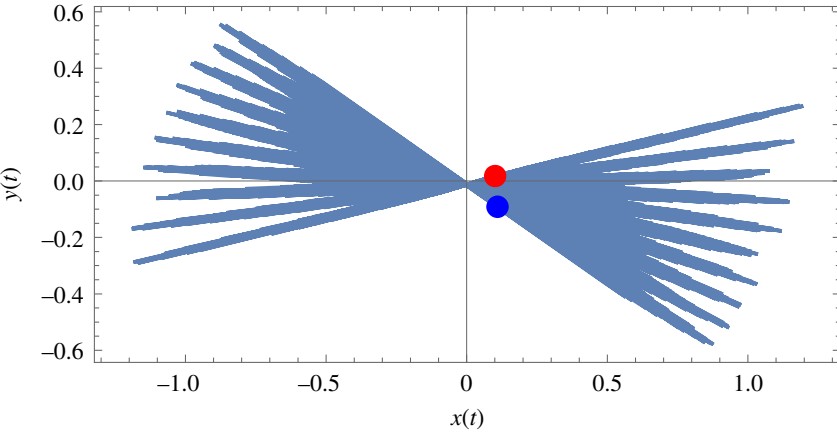

**Figure 3.** Response of the Foucault pendulum to principal parametric excitation with zero detuning and at 0.075 m peak amplitude for a nominal pendulum length of 8 m and a bob mass 2 kg, located in Glasgow (see text above). The red dot denotes the start position, the blue dot the end position. Axes scaled in metres. (Online version in colour.)

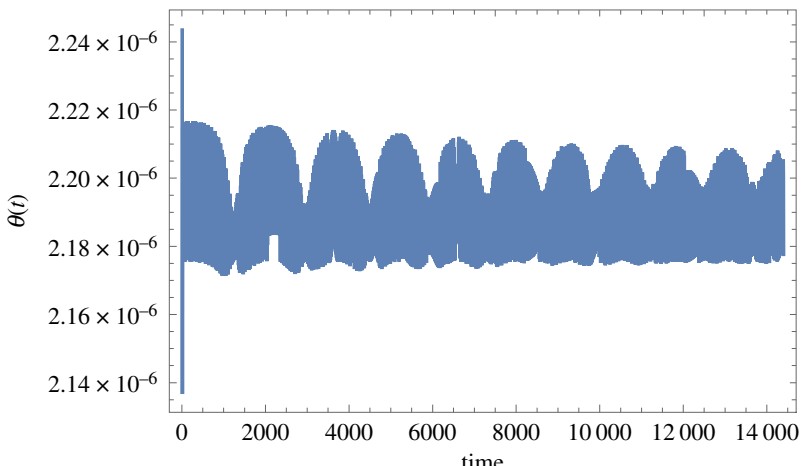

**Figure 4.** Time-domain response of the Foucault pendulum with a 2.54 mm diameter tungsten wire and in pure torsion for the data given in the electronic supplementary material (and also under the same conditions of principal parametric excitation with zero detuning and at 0.075 m peak amplitude, nominal pendulum length 8 m, bob mass 2 kg, located in Glasgow). $\theta(t)$ is in radians and $t$ is in seconds. (Online version in colour.)

again at Glasgow, $r$ is $6363.18 \times 10^3$ m. The mass of the bob $M$ is 2 kg and the radius of the bob $R_{bob}$ is 0.01 m. The density of the air surrounding the pendulum $\rho$ is 1.189 kg m$^{-3}$ and the drag coefficient for the cylindrical bob and locally turbulent air flow $C_D$ is 1.5. Initial conditions for the pendulum are arbitrarily assumed as $x_0 = 0.1$ m, $\dot{x}_0 = 0$ ms$^{-1}$, $y_0 = 0.018$ m, $\dot{y}_0 = 0$ ms$^{-1}$. The pendulum is found to be sensitive to initial displacement conditions only in that they describe a physical starting point, and it is relatively insensitive to the initial velocity conditions, quickly returning to the same time responses from whatever initial velocity conditions are imposed. The integration time chosen for the plots of figures 3–7 inclusive is $t_{end} = 14\,400$ s (4 h), and this is discussed in more detail in §3 as it is shown there to be close to the maximum acceptable limit for numerical integration accuracy when using the MATHEMATICA NDSolve integration routines for this particular series of calculations. Sample code is given in electronic supplementary material.

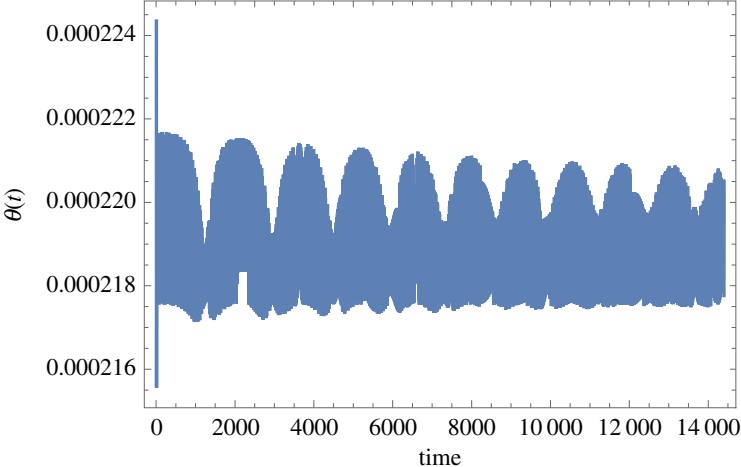

**Figure 5.** Time-domain response of the Foucault pendulum with a 2.54 mm diameter tungsten wire and in pure torsion for $C_B = 0.3$ N m s, the rest of the data as for figure 4 (and also under the same conditions of principal parametric excitation with zero detuning and at 0.075 m peak amplitude, nominal pendulum length 8 m, bob mass 2 kg, located in Glasgow). $\theta(t)$ is in radians and $t$ is in seconds. (Online version in colour.)

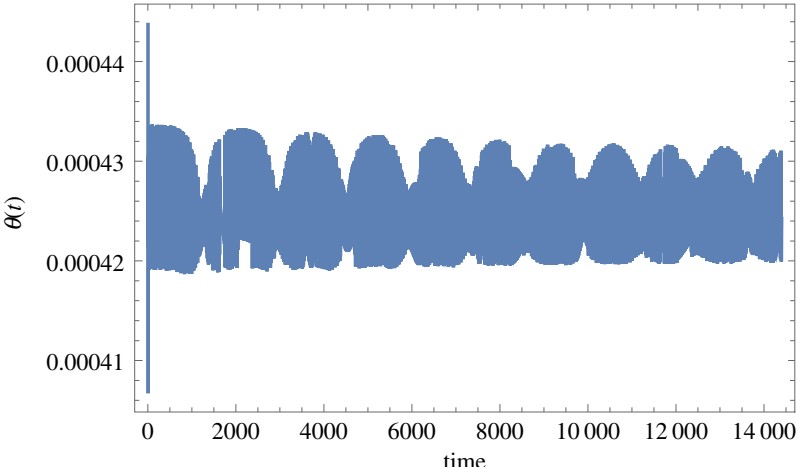

**Figure 6.** Time-domain response of the Foucault pendulum with a 0.813 mm diameter tempered steel wire and in pure torsion for $C_B = 0.003$ N m s, the rest of the data as for figures 4 and 5 (and also under the same conditions of principal parametric excitation with zero detuning and at 0.075 m peak amplitude, nominal pendulum length 8 m, bob mass 2 kg, located in Glasgow). $\theta(t)$ is in radians and $t$ is in seconds. (Online version in colour.)

file ESM2. Accuracy control was implemented as carefully as possible, consistent with eventual convergence time, by setting four principal internal control options for NDSolve stringently: $MaxSteps \rightarrow \infty$, $AccuracyGoal \rightarrow 20$, $PrecisionGoal \rightarrow 20$ and $WorkingPrecision \rightarrow 55$. These choices were based on previous experience with modelling strongly nonlinear systems.

It is also interesting to note that more than one simultaneous resonance could be introduced, potentially increasing the amplification factor even further, and this option is to be vigorously explored in the next phase of the research [22,23].

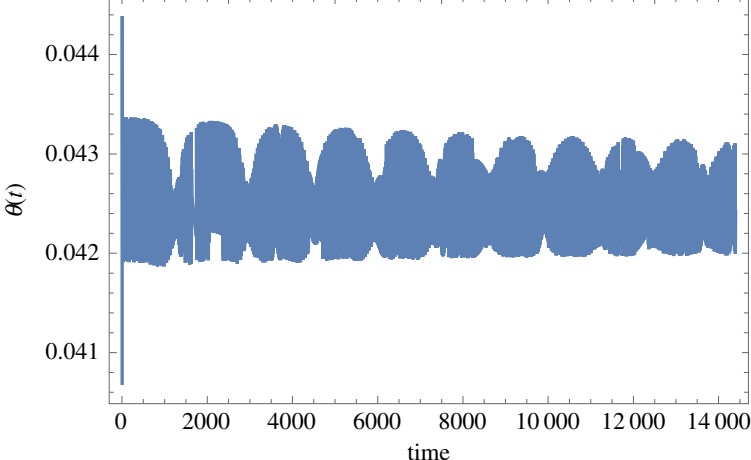

**Figure 7.** Time-domain response of the Foucault pendulum with a 0.813 mm diameter tempered steel wire and in pure torsion for $C_B = 0.3$ N m s, the rest of the data as for figures 4–6 (and also under the same conditions of principal parametric excitation with zero detuning and at 0.075 m peak amplitude, nominal pendulum length 8 m, bob mass 2 kg, located in Glasgow). $\theta(t)$ is in radians and $t$ is in seconds. (Online version in colour.)

## (b) Coupling with pure torsional motion

This analysis has not yet taken into account any possible coupling with unwanted extraneous pendulum modes that might be physically excitable, and because of the design adopted, discussed in §4, the most important of these, over time, is likely to be the pure torsion mode of the pendulum. In principle, such a motion could be excited by a transfer of the small torque across the pivot due to the rotation of the Earth, even in the case of a high-quality spherical bearing, a sophisticated gimballed pivot design, or a precision fluid bearing. This torque would be due to frictional work done within the bearing and proportional to the angular velocity across the joint. An additional governing equation for pure torsional motion is also derived in the associated electronic supplementary material.

The oscillatory torsional response against time, given in figure 4, shows a miniscule peak amplitude, and a tiny DC offset is also predicted, both of which are to be expected given that the excitation is a constant modulated by the parametric excitation of the length $l$. Increasing the spherical rotating joint's damping friction value to $C_B = 0.3$ N m s leads to the time response shown in figure 5, all other data unchanged. It should be noted that $C_B = 0.3$ N m s is an unrealistically high frictional damping value for the spherical rotation joint and yet the torque transfer is still only sufficient to induce a dc offset of around 0.0002 rad, which equates to 0.0114°, for this tungsten wire example. Returning to the manufacturer's stated damping friction of $C_B = 0.003$ N m s then the dc offset reduces to 0.000002 rad, or 0.000114°, as in figure 4, which is 100 times smaller than the value for 0.3 N m s used in figure 5, implying a linear relationship between these two quantities.

If the wire material is changed to tempered steel and the diameter is also reduced notably, so that $S_G = 79 \times 10^9$ Pa and $d_{\text{wire}} = 0.000813$ m, then the torsional response increases in terms of DC offset and amplitude, with the case for $C_B = 0.003$ N m s given in figure 6 and for $C_B = 0.3$ N m s in figure 7.

From the data in figure 6, it can be seen that the DC offset is 0.000426 rad, or 0.0244°, and the peak-to-peak amplitude is approximately 0.000013 rad, or 0.00074°. In the case of the hypothetical case of figure 7 the DC offset is 0.0426 rad, or 2.4408°, and the peak-to-peak amplitude is approximately 0.0013 rad, or 0.07448°. The conclusion from this is that any pure torsional motion due to torque transfer across the spherical rotating joint will be inconsequentially small in the case

of the 2.54 mm diameter tungsten wire. However, if a thinner tempered steel wire of 0.813 mm diameter is used instead then the predicted DC offset approaches 0.025° (figure 6) and it is then just within the bounds of possibility that this could start to exacerbate unwanted couplings over time. In the case of the (unrealistically) high value for $C_B$, explored in figure 7, then this possibility would become much more concerning. Overall, it is safe to conclude that for the design data discussed in §4, the pure torsional mode is unlikely to be excited as a result of torque transfer across the spherical rotating joint due to laboratory rotation, and future design work will ensure that this feature continues to be prioritized.

## 3. Theoretical results for different locations

As shown in figure 3, the analytical model predicts that the Foucault pendulum responds to the rotation of the Earth and the parametric excitation of the length, as one might expect, for the case of a pendulum of a nominal length of 8 m. This analysis can be extended for the same pendulum operated at latitudes from the North Pole down to the Equator and on to the South Pole. The specific effect of latitude on the precession of the pendulum is summarized in table 1 for 11 different locations over a maximum allowable integration time of 14 400 s (4 h) and for the same initial conditions as in figure 3. Assumptions include using specific locational accelerations due to gravity and radii of the Earth, and taking the constant angular rate of the Earth $\Omega$ about the polar axis. The precession $\alpha_{\text{tend}}$ is calculated from

$$\alpha_{\text{tend}} = 57.2958 \left( \arctan \left( \left| \frac{y_0 - y_{\text{cent}}}{x_0} \right| \right) + \arctan \left( \left| \frac{y_{\text{tend}} - y_{\text{cent}}}{x_{\text{tend}}} \right| \right) \right), \tag{3.1}$$

where $\alpha_{\text{tend}}$ is given here in degrees, $x_0$ and $y_0$ are the initial displacement points at $t = 0$ (associated with the red dot in figure 3), $x_{\text{tend}}$ and $y_{\text{tend}}$ are the final displacement points at $t = t_{\text{end}}$ (associated with the blue dot in figure 3) and $y_{\text{cent}}$ is the $y$ value when $x = 0$. Equation (3.1) was computed automatically at the end of each numerical integration in order to obtain the aggregated precession $\alpha_{\text{tend}}$ over the chosen integration time. This precession was then linearly extrapolated over 24 h to give an approximate daily precession in degrees, as follows:

$$\alpha_{24h} = \left( \frac{86\,400}{t_{\text{end}}} \right) \alpha_{\text{tend}}. \tag{3.2}$$

These two quantities are stated in columns 6 and 7 in table 1 for the 11 locations chosen. The final column gives the calculated rates of precession for $t_{\text{end}} = 14\,400$ s at the 11 chosen locations.

The predicted precessions against location given in the sixth column of table 1 for the integration time of 14 400 s (4 h) are proportionally extrapolated to 24 h, using equation (3.2), as shown in the penultimate column, and it can be seen that this approximation shows the predicted precession magnitudes of just over 360° at the poles. This is clearly phenomenologically correct, although the accuracy of the precession is compromised by the simple linear extrapolation of the responses of this nonlinear system.

Plotting the rate of precession due to rotation of the Earth in figure 8 (from the final column of table 1) against latitude shows the distribution of this quantity from north to south, illustrating that it is zero at the equator and maximized at the poles. We can plot the acceleration due to gravity against latitude in figure 9 [24], showing the profile from pole to pole. As a further insight into the effect of location on the radius of the Earth, this quantity is plotted as a function of latitude in figure 10 [25]. Finally, the extrapolated daily precession of the Foucault pendulum due to Earth's rotation against latitude (penultimate column of table 1) is shown graphically in figure 11.

This initial theoretical investigation confirms that the mathematical model of equations (2.1) through to (2.4) provides predictions of the dynamic performance of the Foucault pendulum consistent with expectations, for various locations, and also shows that the effect of parametric excitation, in the form of length modulation, is highly significant, leading to greatly enhanced response amplitude. The numerical integrations in the previous examples were all performed

**Table 1.** Summarizing 11 specific locations from the North Pole to the South Pole.

| location | latitude (°/rad) | radius of Earth (km) | acceleration due to gravity (m s$^{-2}$) | height above sea level (m) | $\alpha_{tend}$ (°) (at $t_{end}$ = 14 400 s) $l$ = 8 m ICs: $x(0)$ = 0.1 m $y(0)$ = 0.018 m $\dot{x}(0)$ = 0 m s$^{-1}$ $\dot{y}(0)$ = 0 m s$^{-1}$ | $\alpha_{24h}$ (°) $l$ = 8 m ICs: $x(0)$ = 0.1 m $y(0)$ = 0.018 m $\dot{x}(0)$ = 0 m s$^{-1}$ $\dot{y}(0)$ = 0 m s$^{-1}$ | Overall rate of precession over $t_{end}$ $\dot{\alpha}_{tend}$ (° s$^{-1}$) $l$ = 8 m ICs: $x(0)$ = 0.1 m $y(0)$ = 0.018 m $\dot{x}(0)$ = 0 m s$^{-1}$ $\dot{y}(0)$ = 0 m s$^{-1}$ |
|---|---|---|---|---|---|---|---|
| North Pole | 90/1.5707 | 6357.00 | 9.8320 | 0.1 | 60.80 | 364.9 | 0.00422 |
| Luleå | 65.585/1.144 | 6360.43 | 9.8235 | 6 | 55.42 | 332.5 | 0.00384 |
| Glasgow | 55.865/0.9750 | 6363.18 | 9.8156 | 40 | 50.38 | 302.3 | 0.00351 |
| Grenoble | 45.166/0.7882 | 6367.64 | 9.8057 | 213 | 42.82 | 256.9 | 0.002973 |
| Addis Ababa | 8.98/0.1567 | 6377.62 | 9.7743 | 2355 | 16.77 | 100.7 | 0.001164 |
| Quito | −0.1807/−0.0031 | 6378.14 | 9.7715 | 2850 | −1.26 | −7.59 | −0.0000875 |
| Darwin | −12.463/−0.2175 | 6377.15 | 9.7826 | 27.8 | −14.34 | −86.1 | −0.0009958 |
| Adelaide | −34.928/−0.6096 | 6371.16 | 9.7971 | 50 | −35.83 | −214.9 | −0.002488 |
| Wellington | −41.28/−0.7205 | 6368.87 | 9.8027 | 31 | −39.82 | −238.9 | −0.002765 |
| Ushuaia | −54.802/−0.9565 | 6363.89 | 9.8145 | 195 | −48.06 | −288.3 | −0.003337 |
| South Pole | −90/−1.5707 | 6357.00 | 9.8320 | 2835 | −60.41 | −362.4 | −0.004195 |

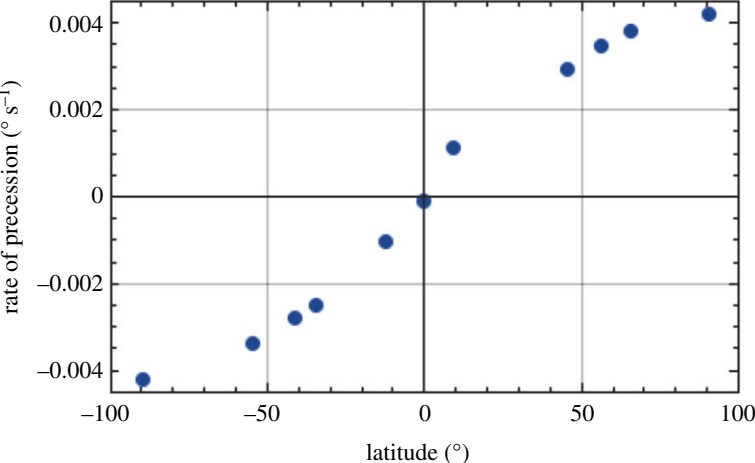

**Figure 8.** Rate of precession due to rotation of Earth, as a function of latitude. (Online version in colour.)

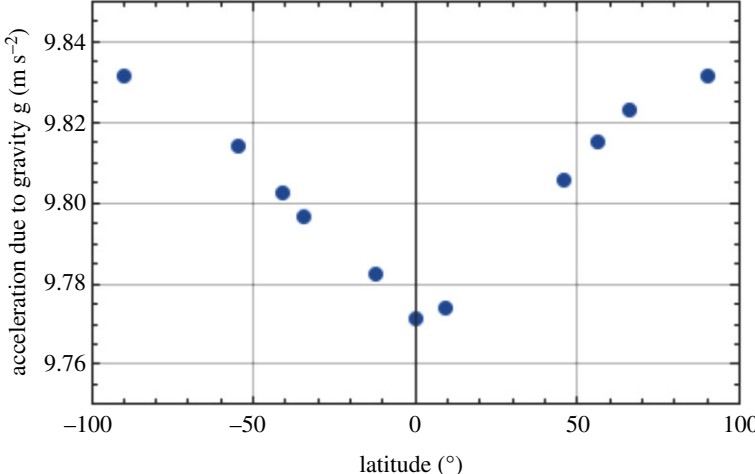

**Figure 9.** Acceleration due to gravity as a function of latitude. (Online version in colour.)

over an integration time of 14 400 s. The next part of this analysis examines comparatively the same cases over reduced run times of 3600 s and 900 s, respectively, and for pendulums of different length. The numerical results given in figure 12 are for pendulums of discrete nominal lengths from 3.5 to 8 m, for three different values of $t_{end}$, and the parametric excitation amplitude $l_1$ set to $0.010l_0$. The aggregated precessions $\alpha_{tend}$ should be constant for each integration time and, therefore, independent of the length of the pendulum, and although this is broadly the case, there are notable discrepancies as the integration time is increased to 14 400 s, particularly for the shorter pendulum lengths. It can be seen that for $t_{end} = 900$ s (grey plot line with triangles), the independence of $\alpha_{tend}$ on length is confirmed and all lengths in the range chosen perform as expected. This is much the same when $t_{end}$ is increased to 3600 s (orange plot line with squares) with only very slightly anomalous behaviour predicted for $l_0$ between 4 and 5 m. However, when $t_{end}$ is increased to 14 400 s (blue plot line with diamonds) then the accuracy of prediction for $\alpha_{tend}$ is seen to have reached its limit, and further numerical tests have unequivocally confirmed that $t_{end}$ values much above 14 400 s give unreliable predictions for the dynamics of the pendulum, irrespective of length. So, $t_{end} = 14\,400$ s has been considered as a strict upper limit on

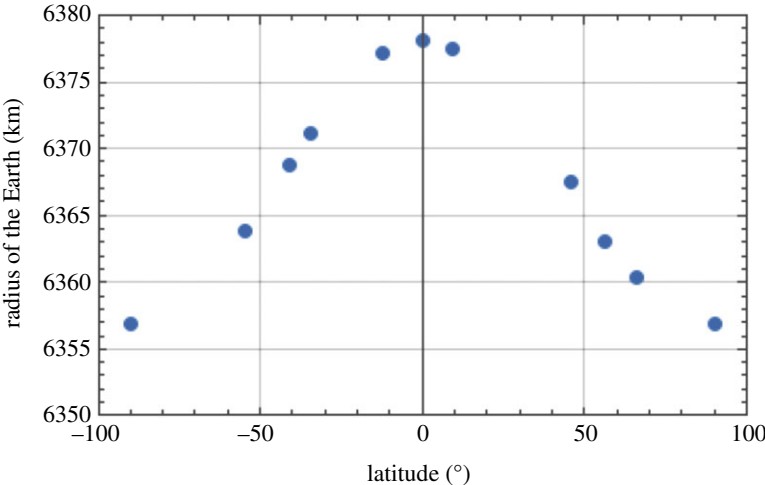

**Figure 10.** Radius of the Earth as a function of latitude (in degrees), showing the oblateness of the Earth. (Online version in colour.)

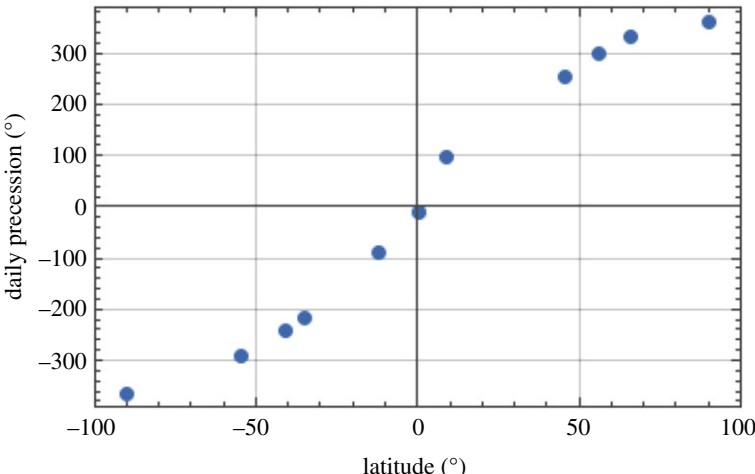

**Figure 11.** Total daily precession due to rotation of the Earth as a function of latitude (in degrees). (Online version in colour.)

acceptable integration time for this study. These trends are virtually identical for two other levels of parametric excitation examined, $l_1 = 0.0075l_0$ and $l_1 = 0.005l_0$, respectively, indicating that the effect of the parametric excitation is as expected, as long as the magnitude of the excitation and the excitation frequency are both maintaining the system within the threshold stability boundary of the region defining principal parametric resonance [19,20]. Values of $l_1$ lower than $0.005l_0$ return back to the threshold stability boundary for this system, and once this is crossed then the effect of the parametric excitation quickly disappears.

This study was further extended for pendulums for which $3.5 \leq l_0 \leq 80$ m, figure 13, and it is interesting to note from these results that the predicted precessions undergo a qualitative change as nominal pendulum length $l_0$ increases from around 8 m for $t_{end} = 900$ s and $t_{end} = 14\,400$ s, but not for $t_{end} = 3600$ s. Further analysis is required to pin down the numerical relationships behind this effect more precisely, so the integration time of $t_{end} = 3600$ s would appear to offer a conservative and reliable basis for the numerical analysis of practically realizable designs

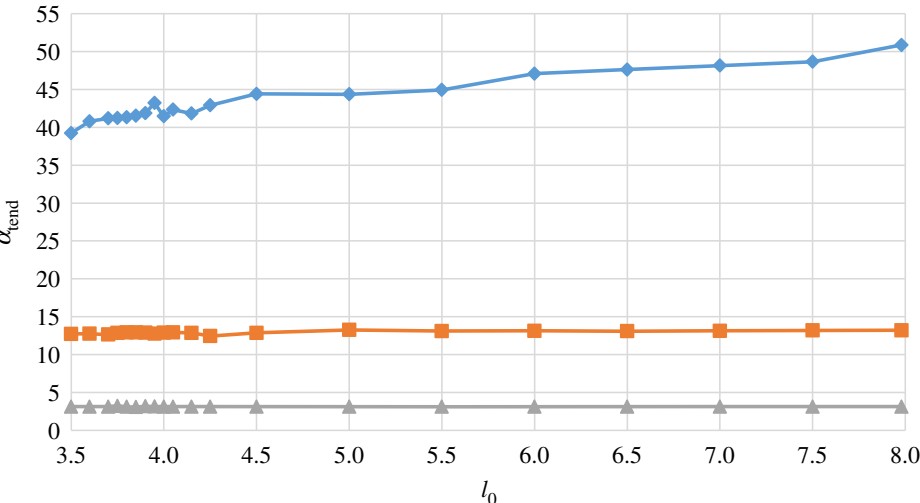

**Figure 12.** Predicted precession $\alpha_{tend}$ (degrees) for $t_{end} = 14\,400$ s (blue line and diamonds), $t_{end} = 3600$ s (orange line and squares) and $t_{end} = 900$ s (grey line and triangles) with $x_0 = 0.5$ m, $y_0 = 0$ m, $\dot{x}_0 = \dot{y}_0 = 0$ ms$^{-1}$, and peak parametric excitation amplitude defined by $l_1 = 0.01l_0$, plotted as a function of nominal pendulum length $l_0$ in metres, for $3.5 \leq l_0 \leq 8.0$ m. (Online version in colour.)

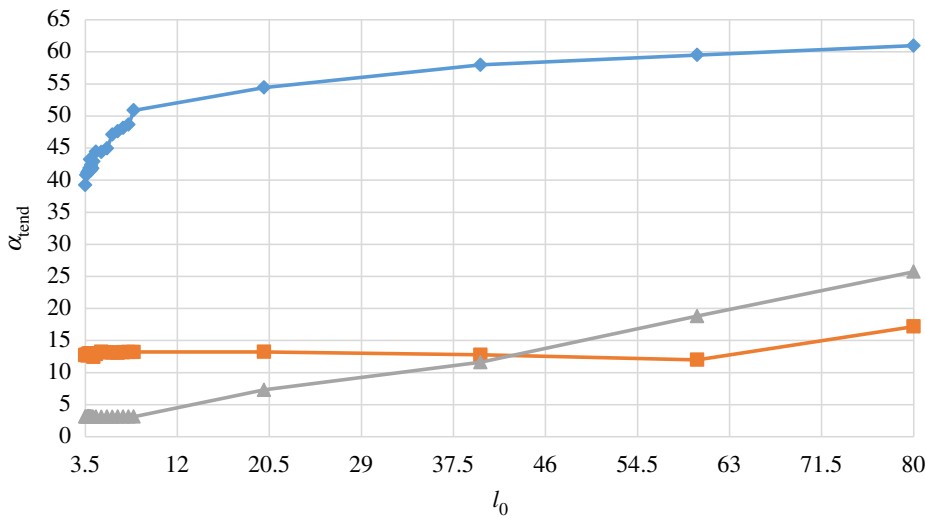

**Figure 13.** Predicted precession $\alpha_{tend}$ (degrees) for $t_{end} = 14\,400$ s (blue line and diamonds), $t_{end} = 3600$ s (orange line and squares) and $t_{end} = 900$ s (grey line and triangles) with $x_0 = 0.5$ m, $y_0 = 0$ m, $\dot{x}_0 = \dot{y}_0 = 0$ ms$^{-1}$, and peak parametric excitation amplitude defined by $l_1 = 0.01l_0$, plotted as a function of nominal pendulum length $l_0$ in metres, for $3.5 \leq l_0 \leq 80$ m. (Online version in colour.)

based around the data discussed so far. This numerical analysis could be further supported by a perturbation analysis using, for example, the perturbation method of multiple scales, on the understanding that the equations would be ordered systematically so that the terms in the differential equations appear at the correct order of perturbation. This could be achieved formally by scaling $\eta$, $\Omega$ and $l$ by means of the notionally small perturbation parameter $\varepsilon$, and usually by non-dimensionalization of the clock time. The generalized coordinates would be approximated

by power series of the form $x(\tau;\varepsilon) = x_0(T_0, T_1) + \varepsilon x_1(T_0, T_1) + \ldots$ and $y(\tau;\varepsilon) = y_0(T_0, T_1) + \varepsilon y_1(T_0, T_1) + \ldots$, where $T_0$ and $T_1$ are successively slower independent timescales, and the two series would be truncated after $O(\varepsilon^1)$ or possibly after $O(\varepsilon^2)$ should there be weaker but physically important nonlinearities that the scheme sets to $O(\varepsilon^2)$. The process is necessarily considerably more complicated if the perturbation analysis is continued to $O(\varepsilon^2)$ [26]. Such a perturbation analysis has not been included in this work to date as the main emphasis has been on a general discussion of the Foucault pendulum, the literature on experimental designs, and a report of our preliminary experiment and some proposals for the next stage of development. It remains to be implemented in a future investigation.

## 4. Experimental design and procedure

The design criteria for the experiment were discussed in the context of the literature in §1 and can be summarized as follows:

1. minimized longitudinal elasticity to mitigate the control of creep over time,
2. maximized symmetry of the upper end suspension,
3. electrically conductive wire to avoid the build-up of static electric charge,
4. homogeneous wire material with minimized residual internal stresses,
5. the longest possible pendulum wire and a long cylindrical bob, preferably both of tungsten,
6. parametric excitation of the length.

Pippard [12] showed that creep could generally be minimized, ideally for tungsten, for the longer pendulum if the wire diameter relates to the bob mass as follows:

$$d_{\text{wire}} \geq 1.8M^{1/2}, \tag{4.1}$$

where the bob mass $M$ is in kg and the wire diameter $d_{\text{wire}}$ is in mm. So, for a bob mass of 2 kg, the corresponding wire diameter should be at least 2.54 mm to guarantee the avoidance of creep. Keeping with the notion of a cylindrical tungsten bob of 2 cm in diameter and $M = 2$ kg then the length required, if made of tungsten, is 32.7 cm. Tungsten parts were duly procured for the wire and the bob to these dimensions. It was attempted to meet the requirement for a symmetrical upper suspension system by installing a specialized spherical rotating joint (*Hephaist* SRJ006C). The parametric excitation was provided by a servo-motor-driven linear ball-screw drive (*Myostat* RD-55 T servo actuator with 12 mm pitch, and 100 mm stroke, and CM1-C-23L20D motor) with a bespoke cylindrical adaptor machined in aluminium alloy to connect the linear drive motor actuator shaft to the spherical rotating joint and then a further cylindrical aluminium adaptor to go from the lower end of the spherical rotating joint to the wire itself. This assembly was intended to provide as symmetrical a suspension as possible; see electronic supplementary material, ESM3 for details (showing the original installation using the tungsten wire). The attachment of the wire to the lower adaptor was carefully considered and it was decided to embed the wire into a 2 cm deep hole in the adaptor, with control of the tolerancing of the hole diameter to provide a suitable location for the use of a very-high-strength epoxy resin adhesive. This proved to be a workable technique and was also used at the lower end of the pendulum for the bob attachment point. On this basis all six design criteria were addressed to a fair extent. A high-roofed laboratory was made available for the duration of the experiment so that the pendulum length could comfortably just exceed 4.5 m. Problems quickly arose with the tungsten wire during installation as the necessities of packaging for transportation of such a long stiff wire introduced severe internal residual stresses which created a static curvature of the wire over the length. This proved almost impossible to eradicate, despite the use of controlled static tension and heat. This material ultimately had to be abandoned in favour of a high-quality tempered steel piano wire of a significantly smaller diameter (0.813 mm) in order to guarantee absolute straightness of the pendulum when suspended, thus not fulfilling Pippard's creep criterion of equation (4.1)

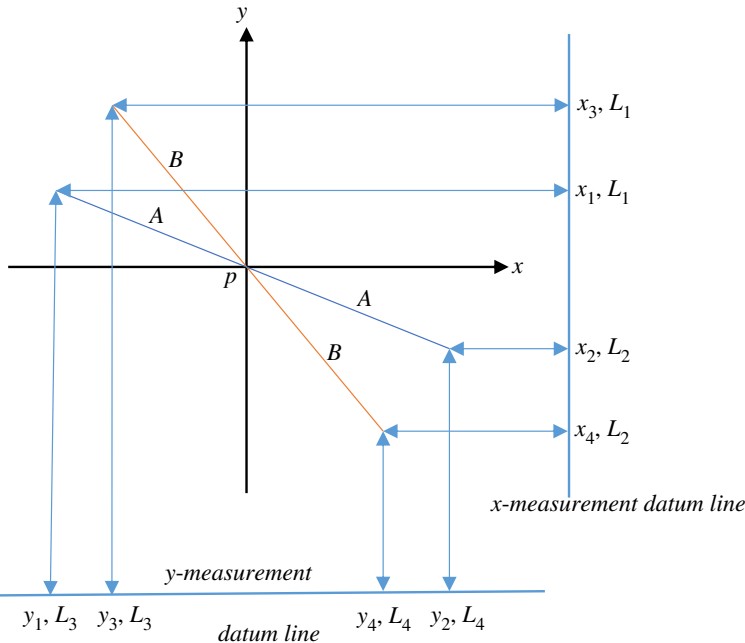

**Figure 14.** Schematic of laser rangefinder layout for the determination of ellipticity. Laser rangefinders $L_{1-4}$ are located on the $x$ and $y$ measurement datum lines to measure distances $x_{1-4}$ and $y_{1-4}$. The initial swing plane is given by the blue line, length $2A$, and the final swing plane is shown by the orange line, length $2B$. Note also that the cross-over point of the swing does not necessarily occur at the origin of the axes. (Online version in colour.)

[12]. New upper and lower adaptors were machined to provide the epoxied attachments for this particular wire diameter. The bob was manufactured from 97% tungsten, and this was successfully used as originally intended. The linear drive unit for the parametric excitation of the suspension assembly was vertically mounted onto a substantial cantilevered channel section, with the actuator protruding downwards through a hole in the section, and the other end of the channel was welded to the upper beam of a large pre-existing structure within the laboratory in order to get the required pendulum length, see electronic supplementary material, file ESM3, figure E3.1 for details. This provided a massive support for the pendulum installation. The system was designed so that the bottom face of the bob was to be 10 cm above the surface of a specially manufactured table, at actuator mid-stroke. The table was used as a perfectly flat area for defining compass-oriented reference axes and also for the location of the laser measurement instrumentation.

The linear actuator servo-motor was driven by a power amplifier located adjacent to the actuator, shown in electronic supplementary material, file ESM3, figure E3.2, and this was driven through a 10 m active repeater USB cable communicating with a host computer on the ground running bespoke software for the control of the actuator amplitude and frequency. The motion of the pendulum was tracked using four laser rangefinders (*Magnusson* IM25 1 mW 635 nm EN60825-1.2014) fitted to precision bench-top tripods and located on the reference table below the pendulum as shown schematically in figure 14, and in practice in electronic supplementary material, file ESM3, figure E3.3.

The principal measurement concerned the ellipticity of the response, and the calculation to test for this was based on the following approach. The rangefinders were initially set up to measure the differences between the local $x$-scale positions of the peak excursions of the pendulum at a location on the wire immediately above the bob, $(x_1 - x_2)$, and the differences between the local $y$-scale positions of the same peak excursions, $(y_1 - y_2)$, and then these measurements were

repeated by moving the lasers along their calibration lines, after a suitable period of time, to give $(x_3 - x_4)$ and $(y_3 - y_4)$. The relationships given in equations (4.2) and (4.3) can then be derived

$$(x_1 - x_2)^2 + (y_1 - y_2)^2 = (2A)^2 \tag{4.2}$$

and

$$(x_3 - x_4)^2 + (y_3 - y_4)^2 = (2B)^2. \tag{4.3}$$

There will be no indicated development of ellipticity for the parametrically excited case over the time period if $A = B$, but if $A \neq B$ then there will be measurable ellipticity for which the length of the semi-major axis is given by $B$. For the unforced case, the condition $A \neq B$ also captures the transient decay of the pendulum's response from the initial conditions over time, and in that case, it is not an absolute indicator of ellipticity, and we return to this in §5a. In order to obtain as much measurement accuracy as possible the four laser tripods were set up in exactly the same manner to provide the measurement system shown in figure 14 and using their in-built bubble levels so that the laser light beams were all set parallel to the table surface. Both pairs of tripods were arranged so that the front two feet of each tripod lay exactly on the $x$ and $y$ measurement datum lines, respectively. A precision square was set up at arbitrary but well-separated reflection test points on the $y$ and $x$ axes, respectively, see figure 14, and initialization measurements from lasers L1 and L2 were recorded, i.e. $x_{10}$ and $x_{20}$, noting that they were very slightly different due to the set-up differences between each laser in the two pairs. Using $x_{10}$ as a reference measurement for laser L1 allowed a suitable correction to be made for L2 so that the actual experimental measurements $x_1$ and $x_2$ could then be correctly recorded, to an aggregated accuracy of $\pm 1$ mm. The same procedure was followed for lasers L3 and L4 to enable measurements $y_1$ and $y_2$ to be made. The two pairs of laser tripods were then carefully moved along the $x$ and $y$ measurement datum lines in order to take the second set of readings, $x_3$, $x_4$, $y_3$, $y_4$. The results of this test are summarized in §5a,b for the parametrically excited case. The pendulum itself was set up as follows. The length of the pendulum from the centre of rotation of the spherical joint to halfway down the bob was measured using a laser rangefinder to be $4685.5 \pm 0.5$ mm (taking into account the accuracy of the digital rangefinder). Calculating the natural frequency of free undamped vibration from $f_n = (1/2\pi)\sqrt{g/l_0} = (1/2\pi)\sqrt{9.8156/(4.6855 \pm 0.0005)}$ Hz leads to $0.230314407 \leq f_n \leq 0.230338986$ Hz. The average of these two calculated values is therefore $f_n = 0.230326696$ Hz (calculated to 9 dp). The pendulum oscillations were observed for 20 swings, and then this observation was repeated, and the overall average period across the two sets of measurements was found to be $T = 4.2260$ s (measured and calculated to 4 dp). This gave the damped natural frequency of free vibration in the fundamental pendulum mode as $f_n = 0.2366$ Hz (also to 4 dp). The difference between the calculated undamped natural frequency and the measured damped natural frequency is $0.006273304$ Hz, recalling the stated difference in accuracy between the two values. In percentage terms, this is approximately 2.72% and in practice, this discrepancy would certainly account for the very small but finite damping inherent in the system. The linear drive actuator was selected to be capable of exciting the pendulum into principal parametric resonance, for which the drive frequency and the natural frequency of the pendulum mode could be related as follows: $\Omega = 2f_n + \varepsilon\sigma$, where $\varepsilon\sigma$ is a small detuning parameter. This required that $\Omega = 0.4732$ Hz, for perfect tuning, and the servo-motor drive software was programmed to generate this frequency exactly. The peak amplitude of the parametric excitation $l_1$ had to be $\geq 0.01l_0$ to be absolutely sure of driving the system into this resonance. This means that $l_1 \geq 46.855$ mm, and so the peak to peak amplitude, $2l_1 \geq 93.71$ mm. The linear drive actuator was supplied as fully rated for 100 mm peak-to-peak amplitude, and was, therefore, run at 100 mm peak-to-peak with $\Omega = 0.4732$ Hz for the parametric excitations tests, summarized in §5b. It is well known that any pendulum which is gravitationally restored is inherently nonlinear [20], with a softening characteristic and three possible solutions (lower and upper solutions stable, middle solution unstable); therefore, the initial displacement condition has to be large enough to allow the parametric resonance to build the response up to the stable upper solution, given that the lower stable solution will be of very small amplitude. Typically,

there can be slow phase interactions within nonlinear parametrically excited systems, which are accentuated when the damping is very low, as it is here [20], noting that the lowest obtained logarithmic decrement value was taken for damping *in situ*. So, if the pendulum is started in such a way that there is a non-zero phase shift between the excitation and the initialized pendulum motion then there will be considerable beating in the response, taking the form of a low-frequency modulation envelope that decays slowly. It was found to be possible to minimize this effect by launching the pendulum so that the phase shift between the parametric excitation and the initialized motion was as close to zero as possible. Trial and error launch tests by hand showed that this was perfectly possible for small pendulum motions (up to 150 mm peak) but much more difficult to achieve in any consistent manner for significantly larger initial displacements. This underpins the need for a mechanized launch system for a future instrumented pendulum in order to provide any chosen initial displacement without introducing any form of additional unwanted motion. To explore this a little further, a series of subjective hand-launch tests was carried out. In the case of the unforced pendulum, it was found that any perceptible initial ellipticity would grow notably with time and would soon completely mask the required planar motion. In the case of this installation, it took around 15 min for the motion to degenerate into an easily recognizable ellipse. This situation was always notably worse when there were observable levels of 'bob-wobble', long-axis torsion, or both present at the launch. Interestingly, it was also noted that the major axis of the response ellipse of the unforced pendulum would orient itself at a shallow re-orientation angle from either the west–east or south–north (local $x$ and $y$) axis lines when launched exactly along either of those axes in a positive direction, and we return to this observation in §§5a and 6, where this angle is defined as $\gamma$. The parametric excitation was then shown to eliminate the ellipticity problem, provided that the launch conditions were reasonably favourable, and with minimized initial phase shift between the excitation and the response. Parametric excitation was also seen to be capable of compensating for, and removing, moderate levels of initial 'bob-wobble' and long-axis torsion, but this was a somewhat subjective observation and more work remains to be done to quantify this aspect. A video of the parametrically excited pendulum operating from 15 min after launch is available as electronic supplementary material file ESM4 and shows the pendulum swinging without any observable ellipticity at all, but still with the modulating phase shift effect alluded to earlier.

## 5. Experimental results

### (a) Transient response case

The transient test data are shown in table 2, measured for the initial condition when the pendulum was launched by hand along the north–south (local $y$) axis and then measured again at exactly 15 min later, and $A$ and $B$ were calculated using equations (4.2) and (4.3). The coordinate system is as shown in figure 14, and the aggregated accuracy of the measurements is ±1 mm.

By means of similar triangles, it can easily be shown that the $y$ coordinate of the cross-over point of the swing at $x=0$ is $443\pm1$ mm. This is the point when the second set of measurements were taken at $t=900$ s. Simple trigonometry can then be used to show that the shallow re-orientation angle $\gamma$ was 11.2° at that moment in time, as stated in table 2. If we then numerically solve equations (2.1) and (2.2) having removed the parametric excitation, for the experimental data, the predicted precession due to Earth rotation for the data of the unforced experimental pendulum over a time period of 900 s is 3.12°. From this experiment, it is obvious that the response of the unforced experimental pendulum was significantly different from the prediction of the mathematical model, with the re-orientation angle being notably larger than the angle of the predicted precession. This indicates that the natural precession due to Earth rotation was overwhelmed by the inherent ellipticity effects which predominate in the unforced experimental system. Table 2 also shows that $B$ is substantially less than $A$ for the unforced case, but it was quite clear from observation that this was due both to the naturally reducing amplitude over the duration of the transient response as well as being a consequence of the observable ellipticity that

**Table 2.** Measurements taken from the transient test.

| initial conditions, at launch, $t = 0$ | after 15 min ($t = 900$ s) |
|---|---|
| $x_1 = 385$ mm | $x_3 = 354$ mm |
| $x_2 = 385$ mm | $x_4 = 434$ mm |
| $y_1 = 730$ mm | $y_3 = 600$ mm |
| $y_2 = 50$ mm | $y_4 = 195$ mm |
| $A = 340$ mm | $B = 206$ mm |
| $\gamma = 0°$ | $\gamma = 11.2°$ |

**Table 3.** Measurements taken from the parametrically excited test.

| initial conditions, at launch, $t = 0$ | after 4 h ($t = 14\,400$ s) |
|---|---|
| $x_1 = 385$ mm | $x_3 = 14$ mm |
| $x_2 = 385$ mm | $x_4 = 756$ mm |
| $y_1 = 730$ mm | $y_3 = 761$ mm |
| $y_2 = 50$ mm | $y_4 = 19$ mm |
| $A = 340$ mm | $B = 525$ mm |
| $\alpha_0 = 0°$ | $\alpha_{tend} = 45°$ |

developed over the time of the experiment. On that basis, any transient test of a pendulum of this construction will necessarily give results emanating from both phenomena combined and so the utility of an unforced pendulum design for any form of accurate measurement is questionable.

## (b) Parametrically excited case

The parametrically excited test data are given in table 3, measured once more for the initial condition at which the pendulum was carefully hand launched exactly along the north–south (local $y$) axis and then measured again 4 h (14 400 s) later, and $A$ and $B$ were again calculated by means of equations (4.2) and (4.3). The coordinate system is as shown in figure 14, and the aggregated accuracy of the measurements is $\pm 1$ mm. There was no discernible ellipticity measured by means of the laser system over the 4 h during which the parametrically excited pendulum was monitored, and so the ellipse re-orientation angle $\gamma$ was undefined when the pendulum was parametrically excited. The modulating phase shift was seen to continue for well over an hour into the run, after which the system gradually settled into a proper steady-state response for which the peak amplitude was defined by $B$ and measured to be 525 mm, noting that this response was bounded principally by the effect of the nonlinear restoring force. The measured pendulum precession was $45 \pm 1°$ during the run time, and this compared well with the theoretically predicted precession for the parametrically excited experimental pendulum of 46.49° found using the numerical simulation based on equations (2.1) and (2.2). This experiment suggests that a parametrically excited Foucault pendulum could be used for the accurate long-term measurement of rotation of the Earth, and furthermore that the inherent problems that traditionally plague Foucault pendulums, of ellipticity, internal and aerodynamic damping, phase shift modulation, and the variability of launch conditions can all be successfully mitigated if the parametric excitation is carefully and accurately implemented.

# 6. Discussion

A mathematical model of the dynamics of the Foucault pendulum has been presented which includes aerodynamic damping with turbulent air flow, parametric excitation of the length, and coupling with pure torsional motion along the long axis. A numerical exploration of the pendulum was undertaken for a range of different parameter values, including different locations. Obvious limitations in numerical accuracy were found for extended integration times, and for the system investigated here it was shown that an integration time of anything much above 3600 s would inevitably lead to an inaccurate solution. Most of the limitations discussed in detail by Pippard [12] were noted in detail and observed, including the presence or otherwise of the mass of the pendulum wire (neglected here but relatively simple to include in the next design iteration), and the need to adhere, if possible, to the creep criterion of [12], in the form of equation (4.1). Importantly, for the data used here, the parametric excitation was seen to amplify the theoretically predicted response of the system significantly but not to affect the precession (as also shown in [12]), and to maintain the accurate operation of the pendulum over time and against the damping as modelled. Possible coupling to long-axis torsional vibration was explored theoretically for two different wire diameters and it was found to be unlikely that this could be excited in any persistent manner in practice.

An experimental system was designed in an attempt to eliminate many of the known inherent problems of the Foucault pendulum, starting with the use of a modern high-quality spherical rotating joint at the support, stiff wire of high quality (both 97% tungsten and tempered steel), and a cylindrical tungsten bob. This close attention to good design principles did not, when under test, remove the well-reported tendency for the unforced Foucault pendulum to display anharmonicity and elliptical motion over time. In mitigation, the design also included a form of parametric length excitation by means of a linear drive actuator at the support, driven by software running on a remote controlling computer. The length excitation was set up to provide sufficient amplitude drive at a controllable frequency in the region of twice the linear natural frequency of free damped vibration of the pendulum to drive it into principal parametric resonance. The intentions behind this have already been discussed and parametric excitation was found largely to work well in practice, notwithstanding low-frequency phase modulation effects which eventually damped out but were also found to be significantly reduced much more quickly by means of a carefully executed launch procedure. Laser instrumentation was set up to monitor the response of the pendulum for both unforced motion (under which it responded purely to an initial displacement condition) and the parametric excitation. It is clear that with care a reliable medium length Foucault pendulum with parametric excitation can be built in the laboratory for the measurement of the Newtonian rotation of the Earth, but it is also very obvious that to do this really accurately will undoubtedly require some important additions to the design, inevitably to include an electromagnetic pusher drive [1,6,7,13,14,17] and a carefully constructed surround to remove air currents, possibly made of Plexiglass and the situation of the pendulum in a seismically quiet environment which is also well away from electromagnetic compatibility (EMC) effects and large ferrous objects. All the findings of the experimental research reported in this paper, and more, would need to be implemented in full if the Foucault pendulum is to be considered as an instrument that could be developed for the more stringent measurements of motions due to general relativity, of which terrestrial measurement of LT precession could be a compelling goal. This possibility has been considered by many, notably [8,10,11,12] and although realization of this is still not close, it is a credible and exciting motivation to consider at the present time. Pippard [12] was not optimistic that it would ever be achievable, whereas Braginsky *et al.* [8] provided a highly detailed specification for achieving terrestrial measurement of LT admittedly based on hugely precise instrumentation, and it is interesting to note that this could now be far more practically attainable than it was when Braginsky *et al.* first proposed it 35 years ago. Pascual-Sánchez [10] and Iorio [11] have also offered encouraging grounds for optimism in their authoritative takes on using highly accurate Foucault pendulum systems for the measurement of LT precession on Earth.

In order to extend the investigation made by Pippard [12] into terrestrial LT measurement, a further analysis into this quantity has recently been completed by Cartmell [27] based on the analogy between Maxwellian electrodynamics and gravitomagnetism, and formulated in notation that is reasonably consistent with the literature. An equation for LT precession at any terrestrial location was derived in [27] as follows:

$$\Omega_{LT} = \frac{0.6632 GM\Omega_{\oplus}}{c^2 R} \cos\theta, \tag{6.1}$$

where $\Omega_{LT}$ is the LT precession, usually conveniently converted to milliarcseconds per year, $G$ is Newton's universal gravitational constant and taken as $6.67408 \times 10^{-11}\,\mathrm{m^3\ kg^{-1}\ s^{-2}}$, $M$ is the mass of the Earth at $5.972 \times 10^{24}\,\mathrm{kg}$, $\Omega_{\oplus}$ is the angular rate of the Earth and equal to $7.2921150 \times 10^{-5}\,\mathrm{rad\ s^{-1}}$, $c$ is the velocity of light which is $2.99792488 \times 10^8\,\mathrm{m\ s^{-1}}$, $R$ is the radius of the Earth which varies with location, and can be taken as $6356 \times 10^3\,\mathrm{m}$ at the North Pole and, for example, $6363.18 \times 10^3\,\mathrm{m}$ at Glasgow in Scotland (and which strictly speaking includes the elevation above the ground of the pendulum bob), $\phi$ is the latitude of the location, and this is $1.5707963\,\mathrm{rad}$ at the North Pole and $0.9750\,\mathrm{rad}$ at Glasgow. Note that in equation (6.1), we give the LT precession as a function of colatitude $\theta$, where $\theta = (\pi/2) - \phi$, and $\phi$ is the latitude as measured from the equator. The other point to mention is that the constant value of 0.6632 accommodates the true radius of gyration of the Earth. Pippard [12] does not explicitly provide an equivalent formula to equation (6.1) but states that the LT precession can be shown to be $220\ \mathrm{mas\ yr^{-1}}$ at the North Pole. Ruggiero & Tartaglia [28] state that the LT precession at the North Pole is $281\ \mathrm{mas\ yr^{-1}}$. Using equation (6.1) and the above data $\Omega_{LT}$ is calculated to be $219.5\ \mathrm{mas\ yr^{-1}}$ at the North Pole, a result virtually identical to Pippard's value. By changing both the latitude and the radius of the Earth at the North Pole to the respective values for the location of Glasgow then the LT precession there can be calculated from equation (6.1) to be $181.5\ \mathrm{mas\ yr^{-1}}$. This is 82.5% of the value of the LT precession available at the North Pole so one might envisage attempting to measure LT at a quiet Scottish location as a significantly cheaper, yet still potentially feasible alternative to doing it at the Pole itself.

# 7. Conclusion

1. A nonlinear mathematical model has been derived for the Foucault pendulum for any terrestrial location, accommodating nonlinear aerodynamic damping, parametric excitation of the length and coupling to pure torsional motion about the long axis of the pendulum. It has been shown both theoretically and experimentally that the parametric excitation can overcome the natural decay in the amplitude due to damping and therefore maintain the response of the pendulum at a constant level over time, and furthermore that the response amplitude can be significantly amplified to a relatively large steady-state value.

2. The potential for pure torsional motion of the pendulum about the long axis has been explored for two different wire types, materials and diameters, and for a physically realistic level and also an unrealistically high level of excitation through torque transfer due to support joint friction, but it has been shown that there is little likelihood of this motion being incited in practice.

3. A numerical study of the Newtonian precession of a candidate pendulum of nominal length 8 m and a peak parametric excitation amplitude of 75 mm was undertaken for 11 different geographical locations from the North Pole down to the equator and on to the South Pole, calculated for 4 h numerical integrations, and then the precessions were extrapolated for 24 h at each location. These results showed, with reasonable approximation, that the pendulum responds proportionally to the Newtonian rotation of the Earth at each location.

4. A further numerical investigation for different nominal pendulum lengths, with the peak parametric excitation amplitude maintained at 0.005, 0.0075 and then 0.01 of the nominal length, revealed that there is a definite upper limit on the acceptable integration time

for reliable results. For the data examined, the safe upper limit to the integration time was found to be restricted to 3600 s or below. It should be noted that this is a specific result for the data used here and that further work remains to be done to generalize this properly, preferably through a comprehensive non-dimensionalization strategy. This will be undertaken in the next phase of the research, along with a perturbation analysis using the method of multiple scales.

5. A parametrically excited experimental pendulum was constructed in the laboratory and several tests were performed using a monitoring system based on calibrated laser instrumentation. The nominal length of the pendulum was 4685.5 mm and the peak amplitude of the parametric excitation was 50 mm, thereby slightly exceeding 1% of the nominal length, and maintaining that relationship (as applied previously in the numerical analyses). The frequency of the parametric excitation applied to the experimental pendulum was set at exactly twice the measured damped natural frequency of free vibration, 0.2366 Hz, in order to stimulate principal parametric resonance.

6. When parametrically excited, and launched carefully from a known initial displacement (zero initial velocity), the pendulum eventually settled into a steady state in the long term. Well-known problems such as ellipticity due to anisotropy, phase modulation, possible forms of Pippard precession and extraneous motions due to imperfect launch (bob-wobble etc.) all disappeared once an observable steady state due to the parametric resonance took over. A confirmatory video of the parametrically resonant pendulum operating steadily 15 min after launch has been made available as electronic supplementary material. The mathematical model predicted the Newtonian precession for the parametrically excited pendulum to within 5%, over 4 h, notwithstanding the limitations of the numerical integration routine when the equations of motion were solved for longer integration times.

7. The experimental pendulum was also tested for zero parametric excitation, therefore with its response simply due to an initial displacement condition, and for all such free transient cases investigated, performance was seen to deteriorate quite quickly, with ellipticity predominating and the major axis persistently rotating away from the initial launch plane through a shallow re-orientation angle. Although this performance was clearly very poor, the amplifying and pervasively corrective influence of the parametric excitation, when it is actuated, means that all future developments should inevitably use that form of excitation as standard, and possibly with more than one simultaneous resonance condition, merely rendering the free pendulum as a curiosity. It would also be advantageous to combine parametric excitation with the use of an electromagnetic pusher coil and to surround the pendulum with an enclosure to remove the unpredictable damping effects of stray air currents which potentially intruded in the experiment discussed here, in order to optimize its performance for measuring Newtonian precession.

8. An application for the Foucault pendulum as an instrument potentially capable of sensing and measuring the gravitational LT precession has also been considered. This study was motivated by earlier work [8,10,11,12] and an expression for terrestrial LT previously obtained by one of the authors [27] was used to calculate this quantity for locations at Glasgow, Scotland and at the North Pole, with figures comparing very favourably with predictions for LT at the North Pole made by the authors in [8,10,11,12]. It is suggested that an experimental design for the construction of a Foucault pendulum capable of making such a sensitive measurement could be conjectured anew, possibly with slightly more confidence than in the past. However, one cannot downplay the extreme performance improvement that will be needed. As a point of comparison, if we calculate the Newtonian precession of the Earth measured over 1 year at the North Pole, this amounts to 131 400°, which equates to $473 \times 10^9$ mas yr$^{-1}$, which compares with 220 mas yr$^{-1}$ for LT according to [12,27]. This suggests an increase in resolution capability of $2.15 \times 10^9$. Similarly, if we compute the annual Newtonian precession of the Earth at

Glasgow, Scotland, this comes to $110\,230°$, which is $396.8 \times 10^9$ mas yr$^{-1}$ (extrapolated from column 7 in table 1). The LT precession accumulation for Glasgow, Scotland, is calculated to be $181.5$ mas yr$^{-1}$ [27], and so the increase in resolution capability for that location is $2.18 \times 10^9$. The relatively small difference between the resolution magnifications at the two locations can mainly be put down to the extrapolation employed to calculate the 24 h precessions shown in column 7 of table 1. The message is clear, we would require an increase in resolution capability of at least $2 \times 10^9$ to obtain a reliable terrestrial detection of LT precession using a Foucault pendulum. This accords almost exactly with the predictions of [10,12]. In order to do this successfully, it would be necessary to use an alternative measurement to be able to remove the Newtonian precession, to exploit amplifying resonance conditions in a long pendulum running over a period of at least 3 years to resolve the measurement accurately over time and to site the experiment at a remote northerly (or southerly) location, in an environmentally protected EMC-free enclosure.

9. While careful design using modern technological solutions such as the spherical rotating joint and materials of exceptional quality will help passively to reduce the problems traditionally associated with Foucault pendulum systems, it remains imperative also to follow the detailed guidance in the literature, much of which has been examined in the preparation of this paper, [1,3,6–8,10–13,17,28] in order to attempt a successful terrestrial detection of LT precession.

Data accessibility. All the data are included within the body of the text and the electronic supplementary material, ESM2.

Authors' contributions. M.P.C. conceived the idea of designing and building a parametrically excited Foucault pendulum of laboratory scale as a basis for the general design of a large-scale installation, and as the precursor to a planned study into the terrestrial measurement of relativistic effects. M.P.C. derived the mathematical model for the parametrically excited Foucault pendulum, performed the first part of the numerical investigation into its dynamic performance and generated figures 3–11 inclusive, tables 1–3 and figures 14 and E3.1–E3.3 in the electronic supplementary material.. He also designed and performed the laboratory experiment and wrote the main text of the paper. J.E.F. and N.A.L. provided significant technical input into the experiment design and discussions of the context for future experimental development of this research. E.H. performed the second part of the numerical investigation into the dynamic performance of the pendulum and was responsible for preparing figures 1 and 2 and generating figures 12 and 13. She prepared some of the content for the introduction section and collated material from the literature which contributed to the discussion in §6. N.A.L. has also developed a fundamental analysis of Lense–Thirring precession which has added greatly to the technical context from which this paper has been written.

Competing interests. We declare we have no competing interests.

Funding. The authors wish to acknowledge the funding made available to them for the experimental work from the Feasibility Study Programme of the Department of Mechanical and Aerospace Engineering at the University of Strathclyde. They also wish to acknowledge the internship made available to E.H. by the University of Strathclyde and the Ecole Nationale Supérieure d'Arts et Métiers de Lille during July and August 2019.

Acknowledgements. The authors wish to thank the following colleagues for many useful and insightful discussions during the course of this initial research programme: Prof. Sheila Rowan, Prof. Sir James Hough and Prof. Martin Hendry of the Institute for Gravitational Research in the School of Physics and Astronomy at the University of Glasgow. The authors would also like to thank Mr Lee Stewart, Mr Alastair Kerr, Mr John Redgate and Mr Drew Irvine of the Department of Mechanical and Aerospace Engineering at the University of Strathclyde for manufacturing parts and installing the experimental pendulum in the laboratory. We also want to thank the anonymous reviewers for their excellent suggestions for improvements to the paper.

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
