## [Reviewer comments · Proceedings. Mathematical, Physical, and Engineering Sciences]

Review History

RSPA-2019-0680.R0 (Original submission)

Review form: Referee 1

Is the manuscript an original and important contribution to its field?

Good

Is the paper of sufficient general interest?

Good

Is the overall quality of the paper suitable?

Good

Can the paper be shortened without overall detriment to the main message?

Yes

Do you think some of the material would be more appropriate as an electronic appendix?

No

Do you have any ethical concerns with this paper?

No

Recommendation?

Accept with minor revision (please list in comments)

Comments to the Author(s)

The paper can be accepted with a few very small changes, they are listed in my report.

Review form: Referee 2

Is the manuscript an original and important contribution to its field?

Marginal

Is the paper of sufficient general interest?

Marginal

Is the overall quality of the paper suitable?

Acceptable

Can the paper be shortened without overall detriment to the main message?

Yes

Do you think some of the material would be more appropriate as an electronic appendix?

No

Do you have any ethical concerns with this paper?

No

Recommendation?

Major revision is needed (please make suggestions in comments)

Comments to the Author(s)

In principle, using a Foucault Pendulum to perform laboratory tests of general relativity is an interesting idea. However, to make this a credible aiming point for this programme of work, the authors must give some idea as to how they will increase the sensitivity by a factor of one thousand million and how they will control the obvious noise sources. Without such an analysis the paper loses its significance.

Review form: Referee 3

Is the manuscript an original and important contribution to its field?

Acceptable

Is the paper of sufficient general interest?

Good

Is the overall quality of the paper suitable?

Acceptable

Can the paper be shortened without overall detriment to the main message?

Yes

Do you think some of the material would be more appropriate as an electronic appendix?

No

Do you have any ethical concerns with this paper?

No

Recommendation?

Major revision is needed (please make suggestions in comments)

Comments to the Author(s)

The authors investigated theoretically and experimentally the nonlinear dynamics of a Foucault pendulum with the aim of confirming insights from the literature on the reliable operation of this device which could be exploited in the future for the measurement of relativistic effects due to terrestrial gravity. Although the paper is interesting and includes experiments to support the simulated model, the authors must perform some modifications by addressing the following comments:

1. In equation (2.4), the authors used a detuning parameter for the excitation of the pendulum around its principal parametric resonance. This can be used when a perturbation technique such as multiple times scales is employed to solve analytically the nonlinear equations of motion. The authors can discuss this point to state clearly if the problem can be solved analytically while using some hypothesis to simplify the nonlinear equations.
2. When Mathematica has been used to solve the nonlinear problem, what is the criteria set by the authors to ensure that the convergence has been reached?
3. What about the stability of the pendulum with respect to the nonlinearity, the parametric excitation amplitude, the damping and the initial conditions? A profound analysis could be carried out or the authors can refer to the literature for their particular case.
4. Is it possible to enhance the stability domain of such a pendulum using simultaneous excitations, which has been proved in the literature for the collective dynamics of coupled pendulums [<https://doi.org/10.1016/j.cnsns.2016.05.012>]? This point should be discussed in the paper.
5. The authors mention in the supplementary material that the other damping is assumed from experience to be equal to 0.0001 Nms. It is not clear why they did not identify it accurately from the experiments. The authors are invited to clarify this point, since a precise value of damping is needed in the model in order to obtain accurate predictions of the dynamic behavior of the pendulum.
6. The quality of several figures should be enhanced.

Decision letter (RSPA-2019-0680.R0)

06-Feb-2020

Dear Professor Cartmell

The Editor of Proceedings A has now received comments from referees on the above paper and would like you to revise it in accordance with their suggestions which can be found below (not including confidential reports to the Editor).

Please submit a copy of your revised paper within four weeks - if we do not hear from you within this time then it will be assumed that the paper has been withdrawn. In exceptional circumstances, extensions may be possible if agreed with the Editorial Office in advance.

Please note that it is the editorial policy of Proceedings A to offer authors one round of revision in

which to address changes requested by referees. If the revisions are not considered satisfactory by the Editor, then the paper will be rejected, and not considered further for publication by the journal. In the event that the author chooses not to address a referee's comments, and no scientific justification is included in their cover letter for this omission, it is at the discretion of the Editor whether to continue considering the manuscript.

- Acknowledgements
- Funding statement

To revise your manuscript, log into <http://mc.manuscriptcentral.com/prsa> and enter your Author Centre, where you will find your manuscript title listed under "Manuscripts with Decisions." Under "Actions," click on "Create a Revision." Your manuscript number has been appended to denote a revision.

You will be unable to make your revisions on the originally submitted version of the manuscript. Instead, revise your manuscript and upload a new version through your Author Centre.

When submitting your revised manuscript, you will be able to respond to the comments made by the referee(s) and upload a file "Response to Referees" in "Section 6 - File Upload". Please use this to document how you have responded to the comments, and the adjustments you have made. In order to expedite the processing of the revised manuscript, please be as specific as possible in your response to the referee(s).

IMPORTANT: Your original files are available to you when you upload your revised manuscript. Please delete any unnecessary previous files before uploading your revised version.

When revising your paper please ensure that it remains under 28 pages long. In addition, any pages over 20 will be subject to a charge (£150 + VAT (where applicable) per page). Your paper has been ESTIMATED to be 28 pages.

Once again, thank you for submitting your manuscript to Proc. R. Soc. A and I look forward to receiving your revision. If you have any questions at all, please do not hesitate to get in touch.

Yours sincerely
Raminder Shergill
proceedingsa@royalsociety.org

on behalf of
Dr Andrew Tolley
Board Member
Proceedings A

Reviewer(s)' Comments to Author:

Referee: 1

Comments to the Author(s)

The paper can be accepted with a few very small changes, they are listed in my report.

Referee: 2

Comments to the Author(s)

In principle, using a Foucault Pendulum to perform laboratory tests of general relativity is an interesting idea. However, to make this a credible aiming point for this programme of work, the

authors must give some idea as to how they will increase the sensitivity by a factor of one thousand million and how they will control the obvious noise sources. Without such an analysis the paper loses its significance.

Referee: 3

Comments to the Author(s)

The authors investigated theoretically and experimentally the nonlinear dynamics of a Foucault pendulum with the aim of confirming insights from the literature on the reliable operation of this device which could be exploited in the future for the measurement of relativistic effects due to terrestrial gravity. Although the paper is interesting and includes experiments to support the simulated model, the authors must perform some modifications by addressing the following comments:

1. In equation (2.4), the authors used a detuning parameter for the excitation of the pendulum around its principal parametric resonance. This can be used when a perturbation technique such as multiple times scales is employed to solve analytically the nonlinear equations of motion. The authors can discuss this point to state clearly if the problem can be solved analytically while using some hypothesis to simplify the nonlinear equations.
2. When Mathematica has been used to solve the nonlinear problem, what is the criteria set by the authors to ensure that the convergence has been reached?
3. What about the stability of the pendulum with respect to the nonlinearity, the parametric excitation amplitude, the damping and the initial conditions? A profound analysis could be carried out or the authors can refer to the literature for their particular case.
4. Is it possible to enhance the stability domain of such a pendulum using simultaneous excitations, which has been proved in the literature for the collective dynamics of coupled pendulums [<https://doi.org/10.1016/j.cnsns.2016.05.012>]? This point should be discussed in the paper.
5. The authors mention in the supplementary material that the other damping is assumed from experience to be equal to 0.0001 Nms. It is not clear why they did not identify it accurately from the experiments. The authors are invited to clarify this point, since a precise value of damping is needed in the model in order to obtain accurate predictions of the dynamic behavior of the pendulum.
6. The quality of several figures should be enhanced.

RSPA-2019-0680.R1 (Revision)

Review form: Referee 1

Is the manuscript an original and important contribution to its field?

Good

Is the paper of sufficient general interest?

Good

Is the overall quality of the paper suitable?

Good

Can the paper be shortened without overall detriment to the main message?

Yes

Do you think some of the material would be more appropriate as an electronic appendix?

No

Do you have any ethical concerns with this paper?

No

Recommendation?

Accept as is

Comments to the Author(s)

This is a revision. The author has fulfilled all my requirements. I'm satisfied

Review form: Referee 2

Is the manuscript an original and important contribution to its field?

Marginal

Is the paper of sufficient general interest?

Acceptable

Is the overall quality of the paper suitable?

Marginal

Can the paper be shortened without overall detriment to the main message?

Yes

Do you have any ethical concerns with this paper?

No

Recommendation?

Accept as is

Comments to the Author(s)

None

Review form: Referee 3

Is the manuscript an original and important contribution to its field?

Acceptable

Is the paper of sufficient general interest?

Good

Is the overall quality of the paper suitable?

Good

Can the paper be shortened without overall detriment to the main message?

Yes

Do you think some of the material would be more appropriate as an electronic appendix?

No

Do you have any ethical concerns with this paper?

No

Recommendation?

Accept as is

Comments to the Author(s)

The authors have addressed my comments sufficiently to recommend publication of the paper in its current form.

Decision letter (RSPA-2019-0680.R1)

Dear Professor Cartmell

On behalf of the Editor, I am pleased to inform you that your manuscript entitled "On the modelling and testing of a laboratory scale Foucault pendulum as a precursor for the design of a high performance measurement instrument" has been accepted in its final form for publication in Proceedings A.

Our Production Office will be in contact with you in due course. You can expect to receive a proof of your article soon. Please contact the office to let us know if you are likely to be away from e-mail in the near future. If you do not notify us and comments are not received within 5 days of sending the proof, we may publish the paper as it stands.

Open access

You are invited to opt for open access, our author pays publishing model. Payment of open access fees will enable your article to be made freely available via the Royal Society website as soon as it is ready for publication. For more information about open access please visit http://royalsocietypublishing.org/site/authors/open_access.xhtml. The open access fee for this journal is £1700/\$2380/€2040 per article. VAT will be charged where applicable.

Note that if you have opted for open access then payment will be required before the article is published – payment instructions will follow shortly. If you wish to opt for open access then please inform the editorial office (proceedingsa@royalsociety.org) as soon as possible.

Your article has been estimated as being 28 pages long. Our Production Office will inform you of the exact length at the proof stage.

Proceedings A levies charges for articles which exceed 20 printed pages. (based upon approximately 540 words or 2 figures per page). Articles exceeding this limit will incur page charges of £150 per page or part page, plus VAT (where applicable).

Under the terms of our licence to publish you may post the author generated postprint (ie. your accepted version not the final typeset version) of your manuscript at any time and this can be made freely available. Postprints can be deposited on a personal or institutional website, or a recognised server/repository. Please note however, that the reporting of postprints is subject to a media embargo, and that the status the manuscript should be made clear. Upon publication of the definitive version on the publisher's site, full details and a link should be added.

You can cite the article in advance of publication using its DOI. The DOI will take the form: 10.1098/rspa.XXXX.YYYY, where XXXX and YYYY are the last 8 digits of your manuscript

number (eg. if your manuscript number is RSPA-2017-1234 the DOI would be 10.1098/rspa.2017.1234).

For tips on promoting your accepted paper see our blog post:
<https://blogs.royalsociety.org/publishing/promoting-your-latest-paper-and-tracking-your-results/>

Thank you for your submission. On behalf of the Editors of the journal, we look forward to your continued contributions to the Journal.

Best wishes
Raminder Shergill,
Proceedings A Editorial Office
proceedingsa@royalsociety.org

Reviewer(s)' Comments to Author:

Referee: 1

Comments to the Author(s)

This is a revision. The author has fulfilled all my requirements. I'm satisfied

Referee: 3

Comments to the Author(s)

The authors have addressed my comments sufficiently to recommend publication of the paper in its current form.

Referee: 2

Comments to the Author(s)

None